# Metallic Nanoparticle-Mediated Immune Cell Regulation and Advanced Cancer Immunotherapy

**DOI:** 10.3390/pharmaceutics13111867

**Published:** 2021-11-04

**Authors:** Adityanarayan Mohapatra, Padmanaban Sathiyamoorthy, In-Kyu Park

**Affiliations:** 1Department of Biomedical Sciences, Chonnam National University Medical School, Hwasun 58128, Korea; mnaditya95@gmail.com (A.M.); ayanaravinth96@gmail.com (P.S.); 2Biomedical Science Graduate Program (BMSGP), Chonnam National University, Hwasun 58128, Korea

**Keywords:** cancer immunotherapy, metallic nanoparticle, tumor microenvironment modulation, immune cell regulation, antitumor immune response

## Abstract

Cancer immunotherapy strategies leveraging the body’s own immune system against cancer cells have gained significant attention due to their remarkable therapeutic efficacy. Several immune therapies have been approved for clinical use while expanding the modalities of cancer therapy. However, they are still not effective in a broad range of cancer patients because of the typical immunosuppressive microenvironment and limited antitumor immunity achieved with the current treatment. Novel approaches, such as nanoparticle-mediated cancer immunotherapies, are being developed to overcome these challenges. Various types of nanoparticles, including liposomal, polymeric, and metallic nanoparticles, are reported for the development of effective cancer therapeutics. Metallic nanoparticles (MNPs) are one of the promising candidates for anticancer therapy due to their unique theranostic properties and are thus explored as both imaging and therapeutic agents. In addition, MNPs offer a dense surface functionalization to target tumor tissue and deliver genetic, therapeutic, and immunomodulatory agents. Furthermore, MNPs interact with the tumor microenvironment (TME) and regulate the levels of tumor hypoxia, glutathione (GSH), and reactive oxygen species (ROS) for remodulation of TME for successful therapy. In this review, we discuss the role of nanoparticles in tumor microenvironment modulation and anticancer therapy. In particular, we evaluated the response of MNP-mediated immune cells, such as dendritic cells, macrophages, T cells and NK cells, against tumor cells and analyzed the role of MNP-based cancer therapies in regulating the immunosuppressive environment.

## 1. Introduction

Cancer is a life-threatening disease characterized by uncontrolled cellular proliferation and metastasis to other organs in advanced stages. It is the second leading cause of fatality worldwide and significantly suppresses the life expectancy of patients [1]. The current first-line cancer treatments include chemotherapy, radiotherapy, and surgery. The effectiveness of these treatments is compromised by various factors. Chemotherapy, which uses chemical drugs to kill cancer cells, is associated with severe side effects because it fails to differentiate between normal and cancer cells, thereby killing healthy immune cells along with cancer cells and damaging major organs. Radiation therapy damages surrounding healthy tissues and often fails to kill all cancer cells in large tumor tissues. Similarly, it is very difficult to surgically excise the complete tumor mass, resulting in tumor recurrence. These therapies fail to control late-stage and metastatic tumors because they migrate to different organs in the body. To overcome these disadvantages associated with the current treatments, nanoparticles have been explored as alternative therapeutic agents to facilitate the efficient accumulation of the target drug in the tumor and enhance the therapeutic outcome [2]. Nanoparticles are very small in size, enabling systemic circulation inside the body and preferential accumulation in the tumor tissue through leaky vasculature. Nanoparticles are functionalized on the surface with various targeting ligands for tumor specificity and facilitated cellular uptake [2]. By reducing the off-target accumulation, the therapeutic drug delivered as a nanoparticulate formulation can minimize the side effects and adjacent healthy tissue damage during cancer treatment. The administration of nanoparticles into organisms can be carried out in many ways. Each route of administration has its advantages and disadvantages. Nanoparticles can be subjected into an organism by multiple routes, such as oral, intramuscular, intravenous, intraperitoneal, and intranasal, which eventually distributes the nanoparticles inside the body. Amongst these, the pulmonary drug delivery system shows a higher degree of drug delivery to the target organ, but on the other hand, it also exhibits potential local toxicity leading to a critical and narrowed use of drug delivery [3,4]. Intravenous and oral drug delivery has better bioavailability and reduced systemic toxicity. Nanoparticles on systemic injection circulate in the blood and accumulate in various organs of the body. The major accumulation of the nanoparticles is traced in the liver. The liver acts as a detoxifying organ that helps in the removal of foreign and hazardous materials out of the system immediately. The nanoparticles are exerted out of the body usually in two ways, such as renal excretion and hepatobiliary ways. The elimination of nanoparticles within a particular timeframe from the body stands out to be one of the crucial necessities for clinical approval. The major driving forces of nanoparticle clearance are the surface chemistry of the particle and the size. Smaller-sized nanoparticles provide better renal clearance. On the other hand, the surface coatings over the nanoparticle, for example, PEG coating, helps in the extended circulation and are always inclined to have a hepatobiliary excretion [5].

Currently, various nanoparticle-based drugs are clinically approved for cancer treatment. NBTXR3/Hensify, developed by Nanobiotix, is a 50 nm hafnium oxide nanoparticle coated with negatively charged phosphate ions that promote the release of significant electrons upon external irradiation [6]. This nanoparticle was approved in 2019 for the treatment of advanced squamous cell carcinoma. Liposomal nanoparticles formulated using a combination of cytarabine and daunorubicin (in 5:1 molar ratio), marketed as VYXEOS, were developed by Jazz Pharmaceutical in 2017 as a treatment against acute myeloid leukemia [6]. However, only a few Food and Drug Administration (FDA)-approved drugs have been marketed as treatments for some cancer types, underscoring the need to develop more effective therapeutics for other cancer types. Numerous liposomal nanoparticle formulations have been clinically investigated in cancer treatment, and many of them are under different phases of clinical and preclinical trials. Liposomal nanoparticles are composed of phospholipids and cholesterols with enhanced drug safety during systemic circulation. However, lipid and polymeric nanoparticles have a lack of long-term in vivo stability during systemic circulation and multifunctional applications. Conversely, metallic nanoparticles (MNPs) are advanced therapeutic materials exhibiting multiple anticancer properties in tumor imaging and therapy, in addition to superior in vivo stability when used therapeutically.

MNPs are multifunctional units with diagnostic, therapeutic, and drug delivery properties that are useful for advanced cancer therapy [7]. The size and shape of MNPs can be precisely controlled and optimized for maximal therapeutic efficacy. Unlike lipid or polymeric nanoparticles, MNPs tend to be highly stable under interstitial fluid pressure [7]. Further, the optical, thermal, electrical, and magnetic properties of MNPs are gaining increased attention in nanoparticle-based cancer therapy. MNPs exhibit a high surface-to-volume ratio, so they can be easily modified on the surface with various drugs, genetic materials, adjuvants, antigens, and targeting ligands to facilitate anticancer therapy. The optical properties of MNPs can be used to intensify in vivo tracking using multiple imaging modalities, such as computed tomography (CT) [8], magnetic resonance imaging (MRI) [9], positron emission tomography (PET) [10], photoacoustic imaging (PA) [11], ultrasound (US) [12] and surface-enhanced Raman scattering (SERS) [13]. Although MNPs are well-known drug carriers, they are also currently being investigated as potential candidates for photothermal therapy (PTT) [14], photodynamic therapy (PDT) [15], sonodynamic therapy (SDT) [16], chemodynamic therapy (CDT) [17], magnetic hyperthermia therapy (MHT) [18], and immune therapy (IT) [19]. MNPs, such as gold nanoparticles (GNPs) [20], copper nanoparticles (CuNPs) [21], and molybdenum oxide nanoparticles [22], exhibit localized surface plasmon resonance and generate local heat upon near-infrared (NIR) laser irradiation. Laser irradiation can also trigger the generation of reactive oxygen species (ROS) in tumors, along with abnormal heat, which induces protein denaturation, cell membrane lysis, cytosol evaporation, lipid peroxidation, apoptosis, and cancer cell death. Similarly, iron oxide nanoparticles (IONPs) are used as MRI contrast agents and in alternative magnetic field-assisted magnetic hyperthermia [18]. Further, ultrasonic-mediated SDT can penetrate deeper tissues and interact with the aqueous environment of the tumor to generate acoustic cavitation, which activates pre-existing sensitizers to produce ROS [23]. MNP-mediated anticancer therapies kill cancer cells and release tumor antigens, which stimulate immune cells to fight against cancer. Treatment with various antigens, adjuvants, tumorigenesis pathway inhibitors, and checkpoint blockers used along with MNP-mediated cancer therapies result in additive, synergistic, and domino effects.

MNP-mediated immunotherapy is classified into MNP-based combination therapies and immune-modulatory agents. MNPs are functionalized with various targeting ligands to target cancer cells and immune cells. MNP-mediated cancer therapies promote immunogenic cell death and tumor antigen release. Thus, the combination of MNPs with immune-modulatory agents, such as adjuvants and TLR agonists, can stimulate the antigen presentation and immune cell activation against cancer cells [24,25,26]. Stealth MNPs have been functionalized with the CD47 antibody and different cell membranes to prevent phagocytosis and promote therapeutic function inside the body [27,28,29]. This review elucidates the role of MNPs in cancer immunotherapy. We discussed the different MNP-mediated immune cells and strategies to modulate TME in order to enhance cancer immunotherapy.

## 2. General Overview of Cancer Immunotherapy

Cancer immunotherapy is currently the burgeoning field of research, as it employs native immune cells for therapy. For years, scientists have been tracing the mechanism by which tumor cells escape the immune reactions, and interesting factors contributing to the immune escape have been documented. Cancer immunotherapy has flourished since the year 1891, when William Bradley Coley injected bacteria into cancer patients to induce an immune response [30]. He injected a streptococcus bacterium to shrink the malignant tumor as a side effect. His idea was eventually tested, and the results revealed that the injection of a streptococcus bacterium decreased the tumor size. The hypothesis of Rudolf Virchow, a German physician who proposed the cell theory, aided the development of an immunotherapeutic approach, which occurred two decades prior (1863) to the development of the first immunotherapy by William Coley. The hypothesis stated that “penetration of immune cells into the tumor happened in the lesions with prolonged inflammation.” The debate converged to a point when Burnet and Thomas proposed a hypothesis on “cancer immunosurveillance” in 1957 [31]. The discovery of dendritic cells (DCs) by R. Steinman and Z. Cohn was a major milestone in the field of immunotherapy. A breakthrough in immunotherapy occurred when Zinkernagel and Doherty [32] reported that the activation of the T-cell immune response against both pathogens and cancer requires the interaction between self-encrypted antigen-presenting molecules [33] and external peptides [34,35]. The first FDA-approved autologous cellular immunotherapy vaccine Sipuleucel-T (Provenge) was heralded as a treatment for patients with minimal or asymptomatic prostate cancer who developed metastasis [36]. This vaccine targets prostate acid phosphatase (PAP), which is overexpressed in the prostate cancer cell surface. The vaccine was developed from autologous DCs activated in vitro using the recombinant fusion protein PA2024, which fuses PAP and recombinant granulocyte-macrophage colony-stimulating factor (GM-CSF) and activates the antigen-presenting cells (APCs), thus inducing an acute immune response.

Immune responses are induced by multiple immune cells that are activated upon stimulation; for example, GM-CSF activates APCs, which in turn activate the T-cell response. Here, APCs are a set of cells that present the surface-modified antigen to immune cells for a specific action. The immune cells eliminate the antigens. DCs are skilled APCs that induce primary immune responses and are responsible for maintaining immunological tolerance to self-antigens and harmless foreign antigens. Ralph investigated the surface proteins of DCs [37] and demonstrated the surface expression of additional major histocompatibility complex (MHC) proteins, which are important factors for antigen presentation to the T cells [30]. The term “T cells” is derived from their point of origin, the thymus. T cells can be both cytotoxic and non-cytotoxic (helper T cells) and aid in the immediate immune response [32]. T cells do not carry a specific antigen; instead, they use T-cell receptors (TCRs) to identify the antigen surface that is presented to them by APCs [38]. Conversely, natural killer cells (NK cells) can identify self-antigens from non-self-antigens. This mechanism reduces the toxicity in native cells. NK cells are a major player in innate and adaptive immunity [33]. NK cells that are in contact with cancer cells release two destructive enzymes (granzymes and perforins), which enable cancer cell death. Finally, macrophages are known for phagocytosis, a process by which the antigens are engulfed by macrophages [39]. Macrophages are involved in tissue development and homeostasis, removal of cellular wastes, pathogen elimination, and regulation of inflammatory response. They are derived from myeloid-derived progenitor cells (MDPCs) in the bone marrow. Macrophage colony-stimulating factors (MCSFs) and granulocyte colony-stimulating factors (GM-CSF) play an essential role in the establishment and regulation of macrophages during the postnatal period. Macrophages are broadly subdivided into M1 and M2 macrophages based on their level of activation. Briefly, M1 macrophages are pro-inflammatory and act on the site of inflammation and clear the antigens, whereas M2 macrophages are anti-inflammatory that reduce the immune response at the inflammation site [35]. Briefly, M1 macrophages facilitate tumor eradication, while M2 macrophages facilitate tumor progression.

## 3. Tumor Microenvironment and Immune Suppressive Cells

The TME acts as a key determinant of tumor formation and progression. It consists of two major components, including cellular and non-cellular components in the extracellular matrix (ECM) [30]. Along with tumor cells, the TME is composed of stromal fibroblasts; immune cells, such as macrophages, lymphocytes, neutrophils, and microglia; and endothelial cells. Collagen, hyaluronan, laminin, and fibronectin are the major non-cellular components of ECM [31]. Both cellular and non-cellular components of the TME maintain cell survival and proliferation via complex signaling networks coordinated by the tumor cells [32]. These complex-signaling networks, involving cellular and non-cellular components, promote uncontrolled cell division and proliferation, leading to hypoxic conditions and thus decreasing the oxygen supply in tumor cells and the TME, which is necessary for cellular activities. Hypoxia leads to the vascularization of blood vessels to further enhance the oxygen content in the TME [33]. This new blood vessel formation in the TME is termed angiogenesis, which is required for cancer cell migration and metastasis.

The TME is responsible for tumor progression and escape of tumor cells from growth suppressors, such as programmed cell death pathways. The TME triggers the doubling of immortal cells and resistance to cell death to sustain the growth of the cells [34]. These characteristics, together with the crosstalk between non-malignant cells and the TME, result in a highly immunosuppressive and multidrug resistant environment [35]. Thus, the non-malignant cells promote tumorigenesis in all stages of cancer, resulting in a highly metastatic cancer [35,36]. In order to achieve this state, it is highly desirable to facilitate the accumulation of fibroblasts in the TME. Usually, fibroblasts are activated during wound healing and repair, aiding in the formation of a new ECM and eventual formation of the structural framework (by collagen) [37,40]. Even after the completion of wound repair, the cancer-associated fibroblasts (CAFs), also termed as myofibroblasts, remain activated. The activated CAFs have a strong impact on tumorigenesis due to angiogenesis [41,42,43], inflammatory cell infiltration, and secretion of growth factors (GFs) and immunosuppressive cytokines [44,45]. The tumor cells secrete both immune-suppressive chemokines, such as IL8, GM-CSF, CCL-5,2,17 and CXCL12,17, and pro-inflammatory chemokines, such as CXCL9 and CXCL10. Due to the secretion of these immune-suppressive chemokines and vascular dysregulation in the TME, tumor-infiltrating lymphocytes (TILs) lose their ability to migrate into the TME and eradicate tumor cells. Auto-inflammation in the tumor is caused by increased metabolic activities, which stimulate the production of epidermal GF receptors, leading to the activation of the NFκB pathway and oncogene mutations [33]. Thus, the innate immune cells fail to kill the cancer cells, resulting in their escape from immune surveillance. This immune suppression is aided by multiple cell types that downregulate the activity of the immune cells and facilitate tumor progression. Some of the cell types that aid in tumor progression are discussed below.

### 3.1. Tolerogenic Dendritic Cells

In the TME, damage-associated molecular patterns (DAMPs) reduce the innate ability of the DCs to mature, thereby decreasing their antigen capturing proficiency and presentation to the lymphocytes. If the DCs exploring the TME region do not mature adequately, they express tolerable levels of tumor antigen, which results in the formation of intolerant T cells. The tolerogenic DCs (tDCs) express CD80 and CD86 (known as costimulatory molecules) at a very low rate but express increased levels of inhibitory molecules, such as programmed death ligand-1 (PD-L1) and CTLA-4 (known as cytotoxic T lymphocyte antigen-4). These tDCs can be induced by various intrinsic factors, including vascular endothelial growth factor (VEGF), cytokines released from tumors, such as interleukin (IL)-10 and transforming growth factor-beta (TGF-β), and other products released from pathogens. IL-10 releases tDCs that activate type 1 regulatory T cells (Treg cells) via immunoglobulin-like transcript 4 [46]. Ran and his co-workers demonstrated that DCs treated with their nanocomplex yielded a better percentage of TGF-β+DCs. TGF-β enhances the conversion of naïve T-cells into Foxp3+ Treg cells, resulting in activity of the DCs, thereby causing a significant reduction in the immune response.

### 3.2. Tumor-Associated Macrophages

Macrophages that infiltrate or are found in the perimeters of the TME are termed tumor-associated macrophages (TAMs) and are associated with the regulation of multiple immune responses [47]. Macrophage formation, migration, and immune suppression processes are discussed in Figure 1. M2 TAMs cause immunosuppression and promote tumor progression by releasing various matrix metalloproteinases (MMP-7 and MMP-9) and increasing the levels of IL-10, VEGF, IL-12, TGF-β and fibroblast GF. The metabolic electron transport chain pathways of M2 TAMs, including glycolysis, the TCA cycle, and the fatty acid synthesis pathway, also undergo severe reprogramming. M2 TAMs upregulate the expression of programmed death-1 (PD-1) and PD-L1. The expression of PD-L1 by TAMs reduces T-cell activation and immune suppression. TAMs facilitate the secretion of IL17 and IL23 cytokines, which promote genetic instability. TAMs also enable the production of cytokines that suppress the T-cell activity in the TME, the upregulation of immunosuppressive surface proteins to escape from immune response, and the production of ROS. They also contribute to the remodeling of the TME by secreting VEGF (angiogenesis) and increasing tumor metastasis in neighboring tissues.

### 3.3. Myeloid-Derived Suppressor Cells

Myeloid-derived suppressor cells (MDSCs) are a group of immature myeloid cells that are capable of inducing multiple mechanisms to suppress antitumor immunity. MDSCs aid in the metastasis and progression of primary tumors. Recent studies reported that the MDSCs profoundly decreased L-arginine levels in cancer by producing Arginase-1 (Arg-1), thus downregulating the activity of T cells at the site of action. Similarly, L-arginine is degraded by inducible nitric oxide synthase 2. MDSCs exhibit increased activity under a hypoxic environment, which is a common feature of solid tumors. MDSCs infiltrate into solid tumors and downregulate T-cell activation. They also prevent T cells from entering the lymph node (LN), where the T cells are activated by tumor antigens. L-selectin (CD62L) transits naïve T cells via the endothelial venules and enables the migration of T cells into the LNs. MDSCs downregulate the expression of L-selectin via proteolytic cleavage from its external domain by expressing ADAM-17 (metalloproteinase 17), which is found in the plasma membranes of MDSCs. The MDSCs also improvise angiogenesis and metastasis by releasing MMP-9 and VEGF in a STAT-3-dependent reaction. MDSCs induce immunosuppression by producing reactive oxygen and nitrogen species and depleting metabolite levels, which are key regulators of T-cell function and stimulate the expression of ectoenzymes that regulate adenosine metabolism. Several studies have also reported that MDSCs secrete TGF-β and IL10, which are directly linked with T-cell-mediated immune suppression and promotion of Treg cell generation. The use of multiple drugs reduces the accumulation of MDSCs; however, none of the therapies completely downregulated the MDSCs owing to their increased heterogeneity. Nevertheless, chemotherapeutic drugs, such as 5-fluorouracil, Docetaxel, and Gemcitabine, decrease cell proliferation and aid in cell apoptosis.

### 3.4. Regulatory T Cells

Treg cells are a subpopulation of T cells that act as immunosuppressors and maintain self-tolerance/autoimmune disorders by restricting T-cell proliferation and cytokine secretion. Treg cells play a major role in the proliferation and metastasis of tumor cells. The C-C chemokine receptor 4 (CCR4) in Treg cells plays a key role in tumor progression and metastasis. The Treg cells display a higher affinity to TcRs than the conventional T cells (Tcon). Nevertheless, both Treg and Tcon cells can detect a wide spectrum of host and non-self-antigens, including pseudo-self-tumor antigens. Treg cells exhibit better costimulatory complexes than the Tcon cells leading to Treg-mediated tolerance due to increased identification and activation of Treg cells. Treg cells not only control Tcon cells but also a wide variety of immune cells, such as B-cells, NK cells, macrophages and DCs, via cell–cell contact and humoral mechanisms. Treg cells are involved in tumor progression by explicitly improving the expression of CTLA4; glucocorticoid-induced TNF receptor (GITR); IL-2, IL-10 and IL-35; lymphocyte-activation gene-3 (LAG3); and TGF-β. Recent studies demonstrated that the expression of Foxp3 in Tcon induces Treg cell-like immune suppression and that all the molecular mechanisms are controlled by Foxp3. Treg cells are found both in the lymphoid and non-lymphoid tissues, including tumors, where they mediate the immune response and reduce inflammation. Treg cells found in non-lymphoid tissues are highly proliferative and predominantly effector Treg cells (CD44-hi and CD62-lo). CCR4 is a marker of Treg cell migration; the cells that lack CCR4 fail to migrate, which in turn leads to severe inflammation at the site of accumulation (predominantly the skin and lungs). Current immunotherapies entail the deletion of overexpressed Treg cells by controlling the large number of molecules available in Treg cells, such as GITR, CTLA4, PD-1 and LAG3. The anti-CTLA4 antibody, which targets CTLA4 expression, is a breakthrough in immunotherapy (shown in Figure 2). The authors of [48] explain the schematic manipulation of immune responses by Treg cells on the T cells. Two of the monoclonal antibodies (ipilimumab and tremelimumab) targeting CTLA4 were tested in patients with prostate, melanoma, and renal cell carcinomas and were approved for use by the FDA [49,50].

## 4. Modulation of the Metallic Nanoparticle-Based Tumor Microenvironment

Metal nanoparticles can improve the TME modulating properties by altering various internal factors, such as ROS production, thermal ablation, glutathione (GSH) levels, and hypoxia, in the TME. The TME is characterized by complex milieus, such as hypoxia, depletion of oxygen at the core of the tumor mass, increased GSH levels to maintain homeostasis as tumor cells exhibit high metabolic rates, and increased ROS levels, which act as a double-edged sword, as they can both promote as well as inhibit tumor progression. These conditions can be altered to destroy the TME and eventually stop tumor progression. Metallic nanoparticles have smaller sizes and possess higher tensile strength for deep tumor penetration alongside the targeting materials. Many metal nanoparticles induce ROS generation and thermal ablation without modification (Table 1 and Figure 3). Thus, metal nanoparticles have gained substantial attention in tumor targeting and eradication strategies.

### 4.1. ROS Generation

Tumor cells exhibit higher metabolic rates than normal cells and produce excessive amounts of ROS in the TME. ROS production is an endogenous process where the cytochrome oxidase receives electrons from membrane carriers to generate superoxide ions. However, the ROS are scavenged by the intracellular GSH, thus maintaining homeostasis. ROS includes superoxide, peroxide, hydroxyl radical, singlet oxygen and alpha oxygen [51]. Elevated ROS production has a huge impact on TME modification [52]. Multiple studies have reported that inflammatory factors, such as cytokines, and growth factors increase the levels of nitrogen oxide required to produce ROS. In this process, IL-20 promotes ROS generation by activating the STAT-3 and protein kinase 3 (AKT) signals. ROS, as effector molecules, attract monocytes to the site of action and further improve inflammation. Thus, ROS generates an inflammatory tumor environment, thereby recruiting suppressor cells into the TME. Increased levels of ROS lead to the activation of p65, TGF-β, and tumor necrosis factor-alpha. In short, increased ROS generation leads to the secretion of IL-1β, IL-8, IL-6, CXCL-12, TNF, NOX2, COX-2 and IL-2 (Figure 4) into the ECM of the tumor, suggesting that ROS stimulates the multiplication of tumor cells and downregulation of all apoptotic factors, thus promoting tumor growth and remodeling the TME for better immunosuppression.

Metal nanoparticles produce ROS via interaction with mitochondria in the cells. MNPs tend to accumulate and depolarize the mitochondrial membrane, thus interfering with the electron transport system of the cell. Interruption of the electron transport system leads to increased levels of the internal singlet oxygen, thus increasing the cellular levels of ROS [53]. In 2018, Chen et al. processed the manganese oxide nanoparticles to evaluate their ability to generate ROS. MnO_2_ NPs were synthesized on the thiol functionalized mesoporous silica NPs (MS@MnO_2_ NPs) through thiol metal interaction. MnO_2_ depletes GSH to GSSG and produces Mn^2+^ ions through a redox reaction. These Mn^2+^ ions react with the intracellular H_2_O_2_ and bicarbonate to produce hydroxyl radicals, which enhance the intracellular production of toxic hydroxyl radicals (·OH). Mn^2+^-mediated GSH depletion prevents OH radical scavenging and promotes enhanced CDT [54]. Similarly, Yang et al. demonstrated that iron oxide induces ROS via the Fenton reaction in the presence of intracellular H_2_O_2_ [17]. However, the intracellular H_2_O_2_ level is not sufficient to process the Fenton reaction and hydroxyl radical generation. A concentration of 20 µg/mL MNPs treatment in the HeLa cell line is insufficient to promote ROS-mediated cell death. Thus, exogenous H_2_O_2_ was required to promote Fenton reaction-mediated CDT. The increased levels of ROS induce the secretion of pro-inflammatory cytokines at both the tumor site and in the systemic circulation. IL-6, IL-1β and TNF-α are secreted at high levels in and around the tumor environment after the exposure of MNPs, which alters the TME and improves immune response. Nanoparticles facilitate the electron transfer process and reduce the mitochondrial membrane potentials, thereby increasing ROS accumulation. Various other metal nanoparticles, such as Fe_3_O_4_, CuO and MnO_2_, can produce a ROS burst inside the TME via Fenton and Fenton-like reactions [55]. MNP-mediated ROS production induces apoptosis and tumor-associated antigen (TAA) release. These TAA productions inside TME promote DC maturation and T cell activation against the tumor. A combination of CTLA-4 blockade therapy controlled the primary and metastatic tumor growth [56].

### 4.2. GSH Depletion

GSH is a triple peptide consisting of cysteine, glycine and glutamic acid and is found in all cells of the human body. GSH helps in scavenging the excess ROS produced by the tumor cells. The concentration of GSH is higher in cancer cells than in normal cells owing to the increased metabolic activity. The synthesis of GSH in cells is a two-step ATP-mediated biosynthesis. The first step is the catalysis of GSH by γ-glutamyl cysteine ligase (GCL), which is composed of two micro-units, namely, GCLC (a highly catalytic unit) and GCLM (a modifier unit). The second step is the catalysis of GSH by GSH synthase. GSH reduces the effects of chemotherapeutic drugs inside the TME, as it scavenges the ROS produced by these drugs for tumor cell apoptosis and the process is highly challenging [57]. However, the role of GSH in immune cell activation and therapy is complex. Usually, the antigens are engulfed by APCs and are digested, and the peptides are presented to the T-helper cells. This presentation to the T-helper cells is facilitated by the MHC-II, which identifies peptides using the surface TCR. In the TME, owing to the high levels of intracellular GSH, the peptides that are targeted for digestion by the APCs carry additional thiol groups. The digestion of thiol-rich cells and their presentation requires MHC-II modifications on the surface and unfolding to reduce the number of thiol groups. The exosome acts as a disulfide-reducing site. Under low GSH levels, the APC only secretes limited amounts of IL-12, thus activating CD2 Th0 to CD4 Th2 cells and eventually generating a cellular response. Conversely, high levels of GSH lead to CD2 Th1 activation and eventually induce a cellular response. When the levels of IL-12 increase, the CD4 Th0 polarizes to CD4Th1, which produces IL-2 and IFN-γ to generate a macrophage-based cellular response.

Tumor cells carry high levels of GSH, as they need to scavenge the ROS that is repeatedly generated by the increased metabolic activity. Excess GSH level in cancer cells scavenges the singlet oxygen and limits the therapeutic efficacy of PDT. MNPs in the TME reduce the concentration of GSH via the formation of a metal-GSH thiolate, which in turn reduces GSH concentration inside the cell [58]. The Cu-TCPP complex enables the reduction of GSH to GSSG (a disulfide inactive form) via a single-step reduction of Cu^2+^ ions to Cu^+^ ions [59]. Cu-TCPP complex depleted the GSH level from 76% to 25% and increased the GSSG level from 24% to 75% in 5 h. ESR spectrum intensity of ^1^O_2_ detection was reduced instantly after the addition of GSH, but it was recovered after 6 h, which confirmed the GSH depletion mediated ROS generation. Intracellular ^1^O_2_ detection was performed through DCFH-DA assay in HeLa cells. The fluorescence intensity of DCFH-DA in HeLa cells was significantly reduced after the addition of the GSH promoter. However, the fluorescence intensity of DCFH-DA was unchanged after Cu-TCPP treatment, which suggests the role of Cu-TCPP-mediated GSH scavenging and ROS production in cancer cells [59]. The decline in GSH concentration of tumor cells in response to the presence of metal nanoparticles induces the generation of excess ROS at the tumor site, thereby increasing inflammation and severe immune response [60,61].

### 4.3. Amelioration of Hypoxia

Hypoxia is a condition where the entire body or a part of the body is deprived of the usual oxygen levels. TME is highly hypoxic due to the reduced blood flow into the core of the tumor region and the lack of oxygen supply. Cancer cells consume a high level of oxygen due to their higher metabolic activity, which reduces the partial pressure of oxygen below 10 mmHg and is known as hypoxia. Hypoxia-inducible factor (HIF)-1α is expressed at the tumor site due to the hypoxic environment and promotes tumor proliferation and migration [62]. Alleviation of hypoxic conditions and HIF1 gene expression can promote tumor therapy. Recent studies reported that targeting the P13K/AKT/mTOR pathways leads to the inhibition of HIF transcription factors. Treatment with an mTOR inhibitor (rapamycin) reduced HIF-1α expression and VEGF levels. Currently, rapamycin and an AKT inhibitor (MK-2206) are used to reduce the expression of PD-L1, thereby improving the immune activities. HIF-1α and VEGF can be passively targeted to improve the immuno-modulation of M1 macrophages. Low doses of anti-VEGF2 antibody enable the polarization of TAMs from M2 (immunosuppressive) to M1 (immunostimulatory) macrophages, thus facilitating the infiltration of CD8^+^ and CD4^+^ cells.

MNPs are utilized in singlet and doublet oxygen generation and deliver the therapeutic loads into the site of action. Metals, such as calcium, iron, copper, manganese, and cerium, can generate intracellular oxygen upon exposure to intracellular peroxides. Tingsheng Lin and his co-workers, in 2018, explained the mechanism of oxygen generation inside the TME (hypoxia) by MnO_2_ NPs. As shown in Figure 4, MnO_2_ reacts with the intracellular H_2_O_2_ to produce oxygen [63]. This catalytic reaction between intracellular H_2_O_2_ and MnO_2_ NPs increases the levels of intracellular oxygen, thus alleviating the hypoxic condition. The dissolved oxygen level was increased with the addition of H_2_O_2_, where a higher concentration of H_2_O_2_ and a longer reaction time between MnO_2_ and H_2_O_2_ produced more oxygen. Similarly, the singlet oxygen production rate was higher in the HSA-MnO_2_-Ce6 + H_2_O_2_ group than the HSA-MnO_2_-Ce6 group. The PDT effect of HSA-MnO_2_-Ce6 was significantly higher than HSA-Ce6. During in vitro cell viability assay, the HSA-MnO_2_-Ce6 group showed an 89.70% cell death compared to a 54.76% cell death in the HSA-Ce6 group. Moreover, in vivo tumor growth was suppressed significantly in the HSA-MnO_2_-Ce6 group due to the enhanced PDT effect [63]. Intracellular oxygen generation by MNPs is an emerging method to alleviate the hypoxic condition, thereby promoting the efficacy of other treatments.

### 4.4. Thermal Ablation

Elevating the temperature level up to 43 °C or above to ablate the tumor is known as hyperthermia. Hyperthermia causes protein denaturation, cell lysis, and cytosol evaporation, resulting in cell death. A specific range of temperatures is selected to avoid necrotic cell death and promote immunogenic cell death (ICD), which attracts antigen-presenting cells inside TME. ICD markers, such as ATP and HMGB-1 and CRT (calreticulin), are increased during the therapeutic intervention and promote antigen presentation to cytotoxic T cells by triggering the release of immune cells. A recent study reported that an optimum temperature is also a key factor contributing to ICD (Figure 5). In this recent study, the tumor was exposed to the following ranges of temperature: 83–83.5, 63.3–66.4 and 50.7–52.7 °C. The results showed that exposure to a temperature of 63.3–66.4 °C significantly induced ICD [64]. The exposure to three different temperatures yielded three different outcomes. At higher temperatures, cell death occurred instantly due to a severe rise in the temperature. A temperature of 63–66 °C resulted in the modulation of TME, induced an immune response, and enhanced the efficiency of thermal ablation. At a temperature of 50–52.7 °C, no notable immune reactions were observed. Conversely, increased temperature eradicates tumor cells and dysregulates the TME to a great extent, enabling the activated T cells to penetrate deep into the tumor sites and eradicate the tumor cells. This also induces the immunotherapeutic response of metastasized cells, as immune cells can trigger ICD in migrated cells. High temperatures lead to non-ICD.

MNPs are vital light-sensitive vectors inducing thermal ablation in tumors upon exposure to NIR laser irradiation [65]. Compared with organic photosensitizers, MNPs tend to exhibit significant thermal stability following multiple irradiations. MNP-mediated thermal ablation involves surface plasmon resonance. Free electrons present on the surface of MNPs are excited and the energy is transferred to a conduction band, which results in photoemission and local heat generation [65]. Multiple metal oxides and metal-hybrid nanoparticles have been reported in laser-irradiated PTT and IT [65].

## 5. Metallic Nanoparticle-Mediated Immune Cell Manipulation

MNPs are unique in modulating the TME and immune cell activities, which synergize the anticancer therapies. By modulating the hypoxic environment to a normoxic condition, MNP NPs inhibit HIF 1 expression, which is responsible for multidrug resistance and therapeutic failure. MNPs mediate GSH depletion, ROS generation, and reprogramming of cold tumors to hot tumors and represent a more easily accessible therapy. MNP-mediated cancer therapies stimulate antigen production and presentation (Figure 6). The reprograming of immune cells against a tumor is an effective way to control tumor growth. MNPs are reported to alter the immune cell types, such as M0/M2 macrophages to M1 macrophages, DC maturation, STING pathway stimulation, NK cell activation and memory T cell production (shown in Table 2). Multiple therapeutic modes are discussed below, based on the type of MNP-mediated immunotherapy.

### 5.1. Metallic Nanoparticle-Mediated Dendritic Cell Maturation

DCs are competent antigen-presenting cells (APCs) of the immune system that drive innate and adaptive immune responses and activate T cells to attack cancer cells [88]. DCs participate in the immune reaction by capturing, processing, and presenting antigens to adaptive immune cells and stimulating their polarization into effector cells [89]. They arise from bone marrow progenitor cells and are classified into conventional DCs (cDCs) and plasmacytoid DCs (pDCs) [90]. DCs contribute to T-cell differentiation and polarization and activate different signaling pathways. The exact mechanisms of Th1 and Th2 immune responses induced by DCs are still debated and may involve antigens, costimulatory factors, and cytokine secretion that affect the polarization of effector T cells. DCs are the most effective APCs and stimulate naïve and effector T cells via antigen cross-presentation. Immature DCs that capture antigens migrate to lymphoid organs and eventually mature, and this process is mediated by chemokine signaling. Mature DCs bind with TCRs and stimulate the release of naïve T cells via antigen-specific T-cell activation. The binding of CD 28 to CD80/CD86 and CD40 to CD40L transmits costimulatory signals to effectively activate the T cells.

Studies recently investigated nanoparticle-mediated DC activation and immune response modulation. Current studies revealed that DCs circulating in the blood and residing in the tissues can capture NPs actively, which is a hot-button topic in studies investigating the use of NPs to control the therapeutic role of DC in cancer [91]. Compared with traditional modulators, NP-mediated delivery of both antigens and adjuvants can effectively turn immature DCs in the lymph organs into mature cells and stimulate T cells to attack cancer cells. MNPs exhibit exceptional DC-targeting activities, as they carry multiple antigens and adjuvants and stimulate the maturation and induction of the release of immune cells to eradicate tumors. The physiological properties of MNPs, such as size, shape, and surface modification, influence DC maturation. Kang et al. reported that GNP-mediated vaccines can be efficiently delivered into the LNs [92]. Hydrodynamic diameters of 10, 22 and 33 nm GNPs conjugated with a model antigen ovalbumin (OVA) altered DC uptake and T-cell cross-priming. Compared with 10 nm GNP vaccines, both 22 and 33 nm GNP vaccines increasingly activate the DC-targeting cells and OVA-specific CD8^+^ T-cells [92]. NanoAu-cocktails, composed of 60 nm GNPs modified with OVA peptides and 80 nm GNP modified with CpG-ODNs, produce pulsed DC and CD8+ T-cell responses [93]. The NanoAu-cocktail is associated with a significant homing ability to lymphoid tissues and improved CCR7 expression in DCs. Further, OVA-specific CD8+ T-cell expression was reported to be 6.5-fold higher in the group treated with the NanoAu-cocktail than in the OVAp and CpG-ODN-pulsed groups. These results specifically elucidated the efficacy of size-dependent MNP antigen and adjuvant delivery to the lymphoid organs and DC maturation [93]. Vang et al. reported that gold nanorods (GNRs), a plasmonic nano vector (PNV), are phagocytized by DCs and promote DC maturation (Figure 7) [94]. A concentration of 200 µg/mL did not affect DC viability, and a concentration of 50 µg/mL was not effective in inducing cellular apoptosis. The expression of CD40, CD86 and MHC class II in PNV-treated DCs was almost similar to that of lipopolysaccharide (a bacterial adjuvant) and induced the maturation of DCs in a time/dose-dependent manner. PNV acted as an adjuvant and upregulated the expression of MHC-II, CD 40 and CD86 [94]. Manganese ions are essential elements for the innate immune system to sensitize the immune cells towards the tumor. Mengze et al. have recently reported that Mn^2+^ in combination with the STING agonist stimulates the cGAS-STING pathway [95]. Mn^2+^ sufficient mice promoted a greater DC maturation, macrophage, and NK cell activation and increased CD8^+^ memory T cells in mice compared to Mn^2+^ insufficient mice [95]. Similarly, Moon et al. have reported that Mn^2+^ coordinated with cyclic dinucleotide (CDN) STING agonist self-assembled into the nanoparticle and enhanced STING delivery [96]. Manganese MNPs augmented STING activity 12 to 77-fold more in human STING haplotypes and promoted STING independent pathways [96].

Evaluation of the catalytic activities of MNPs has paved the way for developing effective treatments. MNPs induce ROS generation, which triggers the production of pro-inflammatory cytokines and ROS scavenging in order to stimulate the release of additional anti-inflammatory cytokines. These cytokines are key signals for DCs to present the antigens and modulate the immune system. Brian et al. reported that TiO_2_ NPs induce DCs to synthesize IL-12 cytokines and polarize T cells toward a Th-1-biased program [97]. By contrast, CeO_2_ NPs secrete Th-2-biased/regulatory cytokines. Oxidative stress induced by TiO_2_ NPs leads to toxicity and inflammation, thereby modulating DC activation and CD4+ T-cell polarization [97]. MNP-based therapies, such as PTT, PDT, SDT and MHT, promote cancer cell death and dying cancer cells release TAAs. Most of the therapies promote ICD in the form of DAMPs, which trigger DC maturation and tumor immunogenicity [98]. The GNR-assisted near-infrared irradiation (ANP) induces ICD by increasing the temperature to 42 °C and releasing ICD markers, such as CRT and HMGB-1 [20]. These ICD markers promote DC maturation. PTT induces DAMPs in melanoma cells in a time-dependent manner following irradiation for 15 min [20]. ANP-mediated photothermal therapy successfully induced ICD and promoted DC maturation (Figure 8) in 38.9% of the irradiated patients compared with that of the controls (13%) [20]. PTT induced the secretion of pro-inflammatory cytokines, which signal immune response. The levels of effector T-cell populations, such as CD8+ and CD4+, were significantly high in the ANP (with laser) group, which strongly established the activation of antitumor immune response in these patients [20]. ANP was combined with HSP-Cas9 for PD-L1 genome editing to enhance the antitumor immune response by inhibiting T-cell silencing via the PD-1/PD-L1 checkpoint pathway. Similarly, the bimetallic Au/Ag nanoparticles irradiated with NIR-II laser can reprogram cold TME to hot TME via PTT and PDT [15]. The combination of PTT and PDT induces ICD in tumors, APC activation, and DC maturation, thereby resulting in the infiltration of T cells in the TME and the reprogramming of cold tumors into hot tumor TM [15]. Au/Ag NRs induce pro-inflammatory cytokine secretion and DC maturation in a dose-dependent manner. Combined with immune checkpoint blockade, Au/Ag NRs induce a strong antitumor response by increasing the expression of CD8+ cells and inhibiting pulmonary metastasis [15]. MNPs show effective APC uptake and are known to promote strong DC maturation and antitumor response.

### 5.2. Metallic Nanoparticle-Mediated Macrophage Polarization

Macrophages reside in all tissues and are the defenders of the innate immune system. They recognize, engulf, and destroy the pathogens in our bodies. Macrophages are classified into two types according to their function. The M1 phenotype represents the classical macrophages involved in pathogen clearance, inflammation, and antitumor immune response, while the M2 phenotype resembles TAMs that perform anti-inflammatory and pro-tumorigenic activities in various cancers, such as breast cancer [99], melanoma [100], ovarian cancer [101], colorectal cancer [102], and pancreatic cancer [103]. M2 macrophages trigger TGF-β secretion, which is responsible for the activation of epithelial-mesenchymal transition via TLR4/IL-10 signaling and cancer metastasis [104]. Th2 cytokines secreted by TAMs negatively regulate the activities of DCs, NK cells and T cells, thereby generating an immunosuppressive TME. Further, TAMs are associated with angiogenic activities, tumor invasion, metastasis, and drug resistance, which downgrade the current therapeutic approaches [39,47,105]. Inhibition and polarization of M2 macrophages to the M1 phenotype in the TME modulate the immunosuppressive nature of and promote effective immune surveillance against the tumor. MNPs are appropriate candidates that can be used to target TAMs.

Tumor hypoxia induces TAMs to generate an instant response to hypoxia-induced gene expression, which promotes therapeutic resistance and tumor progression. Song et al. reported that MnO_2_ NPs exhibit a strong affinity toward oxygen generation via peroxide scavenging, which reverses the normoxic conditions in the TME [106]. HIF-1α and VEGF expression are downregulated by MnO_2_ NPs based on the amount of oxygen generated on-site, thereby promoting hyaluronic acid activity against M2 macrophage polarization [107]. MDSC cell membrane-coated magnetic nanoparticles (MNP@MDSC) exhibit significant immune escape and tumor targeting abilities. MNP-based PTT enhances ICD and promotes the polarization of M2 macrophages to M1 macrophages [107]. Overexposure of iron oxide nanoparticles triggers oxidative stress and DNA damage due to the presence of hydroxyl radicals generated by the Fenton reaction. It generates further Th1 type cytokines and polarizes M2 to M1 macrophages in order to initiate T-cell activity against tumor metastasis [108]. Chen et al. reported that iron oxide NPs embedded with mesoporous silica nanoparticles (IO-LPMONs) preferably stimulate macrophage polarization and T-cell activation, thereby promoting tumor suppression [109]. The large pore size of mesoporous silica contributes to the antigen payload, which results in DC maturation, and iron oxide NPs induce the polarization of tumor killing M1 macrophages (Figure 9). In another study, iron oxide NPs were treated with RAW 264.7 cells to investigate the programing of tumoricidal macrophages. The results indicated that treatment with iron oxide NPs induced M1-based mRNA expression. Iron oxide NPs induced the release of pro-inflammatory cytokines and activated the CD4+ and CD8+ T cells, which dominated the immunosuppressive growth environment [106]. The surface charge of super-paramagnetic iron oxide nanoparticles (SPIONs) played a crucial role in the induction of M1 macrophages [110]. Positively charged SPIONs (S+) show a high cellular uptake due to electrostatic interaction and high macrophage polarization. Interestingly, negatively charged SPIONs (S−) show preferable cellular uptake and macrophage polarization, whereas neutrally charged SPIONs show minimal cell uptake and polarization in tumor cells [110]. Zhengying et al. reported that magnetite IONPs contain a stimulated interferon regulatory factor 5 signaling pathway and downregulate the M2-associated arginase-1 (Arg-1) expression, thereby polarizing M2 macrophages to M1 macrophages (shown in Figure 10) [111]. Both magnetite IONPs (Fe_3_O_4_@D-SiO_2_) and hematite (Fe_2_O_3_@D-SiO_2_) coated with dendritic silica cells showed differential iron accumulation in cell uptake studies. Fe_3_O_4_@D-SiO_2_ NPs show a significantly high iron content compared with Fe_2_O_3_@D-SiO_2_ NPs due to the difference in intrinsic structures, which controls iron leaching in cells, magnetism-induced aggregation, and oxidation of IONPs into ions that affect Fe^2+^ and Fe^3+^ levels. Pro-inflammatory signaling pathways, such as NF-κB and IRF5, were activated by Fe_3_O_4_@D-SiO_2_ NP treatment and showed M2 to M1 macrophage polarization [111]. Fe_3_O_4_@D-SiO_2_ NPs increased intracellular ROS levels and activated the NF-κB pathway, which was specifically dependent on IRF5 activation [111]. MNPs exhibit a range of physiological properties that facilitate the polarization of the macrophage phenotypes in TMEs in anticancer immunotherapy.

### 5.3. Metallic Nanoparticle-Mediated T-Cell Stimulation

Effective programming of cytotoxic T cells against tumors is the ultimate strategy in cancer immunotherapy. MNP-based modulation of TME can control the immune response and T-cell activation. TiO_2_ NPs alter the TME by secreting inflammatory cytokines, which enhance the maturation of DCs and the activation of T cells [112]. IONPs are adjuvant subunit vaccines, which are administered via intranasal and subcutaneous injection to elicit Th1 and Th17 immune response. They promote the expression of CD86 in DCs and induce the synthesis of pro-inflammatory cytokines, such as IL-6, TNF-α, IL-1β, IFN-γ and IL-12, which are associated with Th1 (IL-12) and Th17 (IL-6 and IL-1β) response [113]. In addition, immune cells, such as DCs, macrophages and T cells, are sensitive to IONPs. OVA-modified IONPs scavenge GSH levels and enrich the production of IL-2, Il-4 and IFN-γ, which are markers of Th-1 and Th-2 responses [113]. Combination therapies can bypass the immune system by inducing ICD-mediated DC maturation and T-cell activation. Currently, ICD-mediated T-cell activation with combination anticancer therapy is a hot research topic. The combination of PTT and IT triggers tumor ablation and ICD to generate an immune response. Zhou et al. reported that manganese ferrite (MnFe_2_O_4_) modified with OVA and R837 synergizes PTT and IT against breast cancer [19]. MnFe_2_O_4_-based laser irradiation-induced thermal ablation triggered the release of R837 and TAA inside the tumor. The release of TAA and R837 stimulated DC maturation and activated T cells. An increase in the iron content of MnFe_2_O_4_ NPs downregulated IL-10 production and M2 macrophage polarization. The PEG-MnFe_2_O_4_ NP-based PTT group showed more significant DC maturation than the control groups; the proportion of TNF-γ-secreting CD3^+^CD4^+^ cells in the group treated with a combination of R837-OVA-PEG-MnFe_2_O_4_-based PTT was up to 1.16 ± 0.07%, and the proportion of CD3^+^CD8^+^T-cells was 1.13 ± 0.17% [19]. Combination therapy inhibited the occurrence of lung metastasis and adjoining tumor tissue. Similarly, a GNR modified with bovine serum albumin, and the immunoadjuvant R837 (*m*PEG-GNRs@BSA/R837) was developed for melanoma treatment by combining PTT and immunotherapy [14]. Laser irradiation-induced thermal ablation enhanced TAA production and adjuvant release, which were taken up by APCs for effector T-cell production and the generation of strong antitumor immunity [14]. Combination therapy was associated with a significant survival rate and metastasis inhibition. Further checkpoint blockade therapy balances tumor-mediated immunosuppression [14]. Exposure to iron oxide nanoring-based mild magnetic hyperthermia (temperature: 43–44 °C) caused CRT expression, which acted as a phagocytotic signal transmitted to the immune cells. Mild thermal ablation induced an 88% increase in CD8+ T-cell infiltration during the treatment of primary and distant tumors [18]. Thermal ablation combined with PD-1/PD-L1 checkpoint blockade therapy increased the CD8+ T cell population and decreased that of the MDSCs [18]. Manganese MNPs have additional potential to augment the STING pathways and promote CD8+ T cells and memory T cells. Moon et al. have reported recently that Mn^2+^ coordinated CDN NPs amplified the STING activation compared to other metals. The combination of Mn^2+^ and STING agonists enhanced the T cells population in mice and suppressed the tumor growth [96].

### 5.4. Metallic Nanoparticle-Mediated NK Cell Delivery

NK cells are skilled immune effector cells that play a crucial role in immunotherapy. They are an independent arm of the immune system and are involved in receptor-mediated activation of targeted cells. NK cells can directly eliminate cancer cells, thereby avoiding factors interfering with antigen presentation and T-cell activation. After activation, NK cells modulate the adaptive immune system by releasing chemokines, such as IFN-γ and TNF-α. According to clinical studies, a high degree of NK cell tumor infiltration is associated with improved cancer prognosis [114]; NK cells can kill cancer cells by releasing perforin and granzymes, which lyse the tumor cells. MNPs show multiple functions, such as delivering payloads, interacting with cellular functionality, and imaging, which enhances NK cell activation and expansion. Jang et al. reported that iron oxide nanoparticles (Fe_3_O_4_/SiO_2_) can be used to target the infiltration of NK-92MI cells in the tumor site via an external magnetic field (shown in Figure 11) [115]. These results indicated that MNP-loaded nanoparticles are effective and safe delivery agents that carry immune cells to the tumor site under an external magnetic force [115]. Similarly, Liya et al. reported the magnetic delivery of iron oxide nanoparticle-loaded NK cells for anticancer therapy [116]. Iron oxide nanoparticle loading did not affect the biological function of the NK cells. The external magnetic field recruited a significant number of CD56^+^ NK cells in the tumor and controlled the immune system [116]. The differences in the size of GNPs influenced cell uptake, the proliferation of NK cells, and pro-inflammatory cytokine production. Compared with 2 nm GNPs, 12 nm GNPs showed a significant increase in CD56+ NK cell population and IFN-gamma cytokine and granzyme B production [117].

## 6. Limitations and Future Perspectives

The application of MNPs in drug delivery and tumor therapy has been investigated intensively. The application of MNPs in tumor therapy is limited by their toxicity and the rate of accumulation and excretion from the body [125,126]. Biodegradation and biodistribution inside the body are the primary concerns limiting the use of MNPs, suggesting the need for further investigation in the future. Several studies reported the incidence of acute and chronic toxicities using metals for therapeutic purposes. The synthesis and surface modification of MNPs must be carried out to avoid these limitations before contemplating a clinical application. Similarly, naked MNPs exhibit immune response by releasing immune cells, thereby compromising the targeted efficacy. Green synthesis, the process by which nanoparticles are produced using natural resources, such as bacteria, plant extracts and fungi, is performed to produce various MNPs to prevent toxicity, which is usually attributed to chemical synthesis [127]. Biosynthesized MNPs show limited protein corona formation in the blood serum, which is a great option for future research [128,129]. Surface modification of MNPs is suggested to avoid the aforementioned shortcomings. Thus, MNPs need to be surface modified to increase the circulation time in vivo. PEG coating over the surface of the nanoparticles was found to increase the stealth life of the nanoformulation for in vivo treatment [130,131]. In addition to PEG, a variety of polymers, namely, polyoxazolines, *N*-(2-hydroxypropyl) methacrylamide, polybetaines, polyglycerols and polysaccharides, facilitate the increased circulation of the nanoformulation [27,132]. Although surface coating increases the circulation of nanoformulations, it cannot protect the nanoformulation completely from immune response and cannot ensure delivery to the target site within the minimum time frame. Hence, metallic nanoparticles are no longer used combined with other therapies in clinical trials. Current developments in utilizing biomineralized nanoformulation for sustained and multifaced therapy have opened the gates to the incorporation of new and efficient drug delivery systems for a wide variety of cancer types [133,134]. Biomineralized encapsulation of metal nanoformulations prolongs the half-life in the blood circulation and prevents protein corona formation from reducing the immune response before reaching the target. For example, BSA, which shows better biodistribution and relatively less toxicity, is currently employed in cancer nano-theranostics [135,136]. Proper tuning of the nanoformulation reduces its accumulation in the liver and spleen and facilitates its delivery to the tumor area, enabling better tumor targeting and eventually reducing the side effects caused by the accumulation of these materials in normal tissues. Future studies should focus on completely overcoming the immune response, for which metal nanocarriers are one of the most appropriate carriers for targeted therapy against a wide variety of cancers.

## 7. Conclusions

MNPs represent an effective therapeutic nano agent for the delivery of various immunomodulatory components, such as antigens, adjuvants, cytokines, and checkpoint inhibitors. MNPs trigger tumor antigen release following multiple tumor therapies and induce a strong immune response. MNP-mediated immune cell modulation, such as DC maturation, TAM polarization, and T-cell activation, evoke an immunotherapeutic response against cancer. Iron oxide nanoparticle-loaded multiple immune cells, such as T cells, macrophages, and NK cells, are delivered to the tumor by targeting an external magnetic field. MNP-loaded immune cell delivery to the TME effectively increases the immune cell population and triggers anticancer immunotherapy. MNPs can repolarize TAMs and immature DCs to M1 macrophages and mature DCs, respectively, which is beneficial for future cancer therapies. Nevertheless, most of the studies investigating MNPs are still in the preclinical stage. The lack of clear evidence supporting MNP-based immune cell response is a downside. However, the unique properties of MNPs leverage the importance of MNPs in cancer therapy, and intensive studies can lead to the application of MNPs in cancer immunotherapy.

## Figures and Tables

**Figure 1 pharmaceutics-13-01867-f001:**
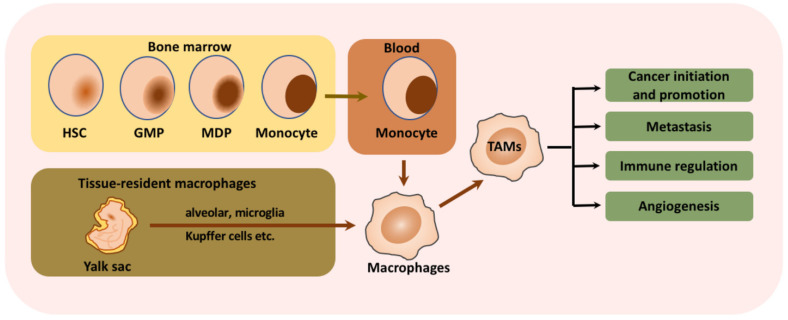
Schematic illustration of the origin of tumor-associated macrophages and their functions. Macrophages are derived from hematopoietic stem cells (HSCs), which are differentiated into granulocyte-macrophage progenitors and further into monocyte-dendritic cell progenitors. Other than HSCs, tissue-resident macrophages are key sources of macrophages that differentiate in situ into alveolar macrophages, Kupffer cells, and brain macrophages. These monocytes released into the bloodstream are activated in the tumor microenvironment and undergo drastic changes that facilitate tumor proliferation, metastasis, angiogenesis and immune modulation. Reproduced from Lin et al. [47] which is licensed under a Creative Commons Attribution-(CC BY 4.0) International License (http://creativecommons.org/licenses/by/4.0/).

**Figure 2 pharmaceutics-13-01867-f002:**
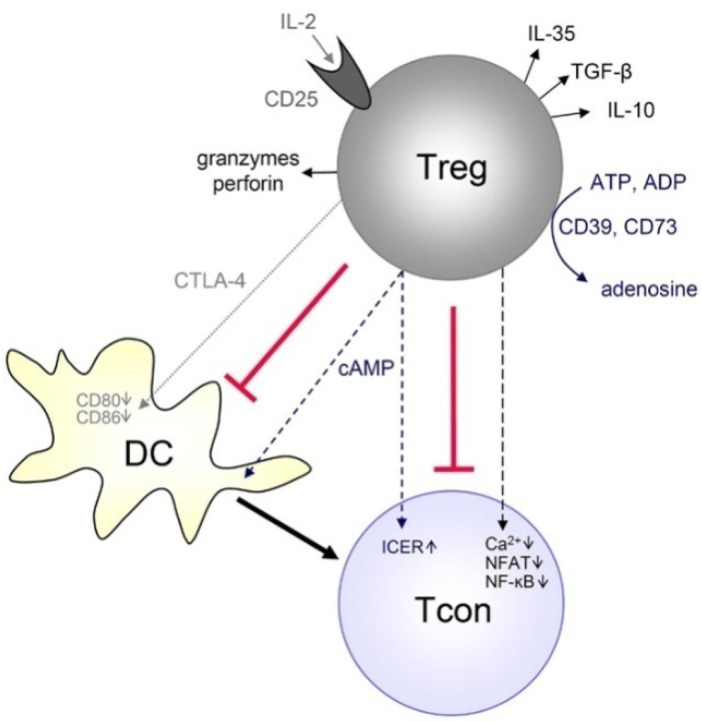
Activation of CTLA-4 in Treg cells leads to the inactivation of the immune response. αCTLA-4 can be used to mask the activity of Treg cells, thus enhancing the T cell binding ability and activation for tumor eradication. Reproduced from Aboulkheyr Es et al. [45] which is licensed under a Creative Commons Attribution-(CC BY 4.0) International License (http://creativecommons.org/licenses/by/4.0/).

**Figure 3 pharmaceutics-13-01867-f003:**
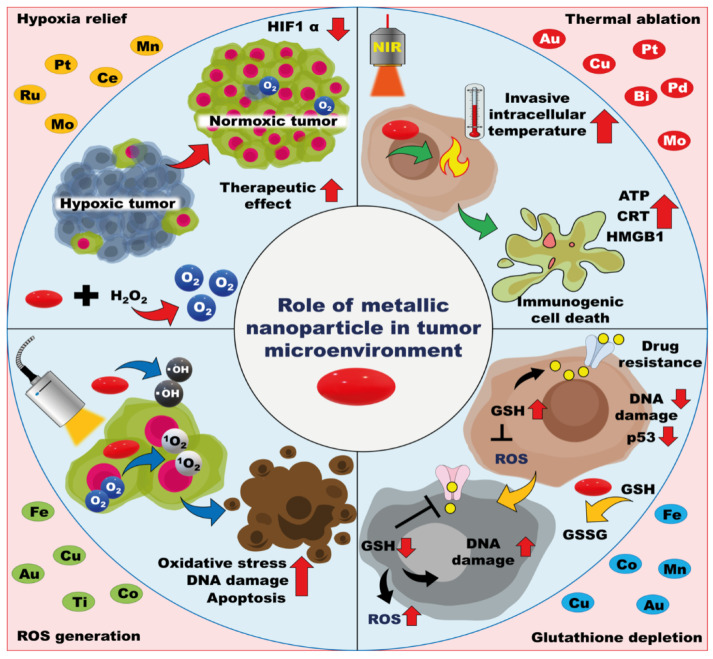
MNP-mediated TME modulation. MNPs alleviate tumor hypoxia by generating sufficient oxygen supply, thus promoting therapeutic activities. MNP-mediated GSH depletion inside cancer cells enhances the expression of excessive ROS and causes tumor cell death. Thermal ablation and ROS production increase temperature and induce oxidative stress, which leads to immunogenic cell death (ICD).

**Figure 4 pharmaceutics-13-01867-f004:**
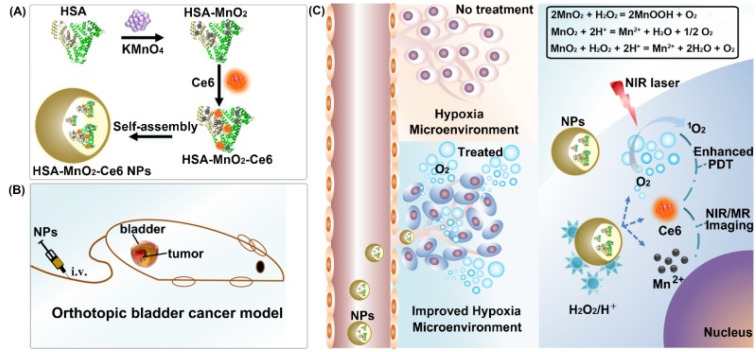
Scheme representing the production of oxygen from H_2_O_2_ under laser irradiation, thus relieving hypoxia in the TME. (**A**) Formation of the nanocomplex, (**B**) intravenous injection of the nanoformulation and (**C**) mechanism of nanoformulation action inside the cell following combination therapy wherein Ce6 increases the efficiency of PDT and helps in imaging, Mn^2+^ increases ROS production, and oxygen is generated via reaction between intracellular H_2_O_2_ and hydrogen. Reproduced from Wu et al. [60] which is licensed under a Creative Commons Attribution-(CC BY 4.0) International License (http://creativecommons.org/licenses/by/4.0/).

**Figure 5 pharmaceutics-13-01867-f005:**
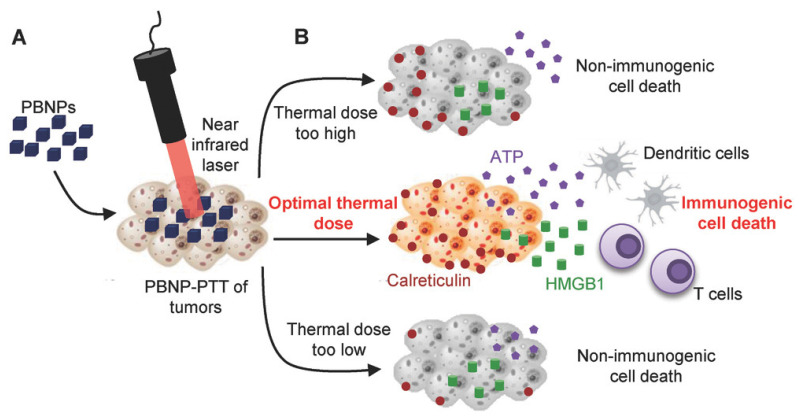
PTT-based thermal ablation of tumor cells, causing tumor cell death and inducing multiple immune responses at different temperatures. (A) NIR laser irradiated Prussian blue nanoparticles for photothermal therapy. (B) Role of thermal doses in cancer cell death. Optimal thermal dose promotes immunogenic cell death and immunotherapy. Reproduced from Sweeney et al. [64] which is licensed under a Creative Commons Attribution-(CC BY 4.0) International License (http://creativecommons.org/licenses/by/4.0/).

**Figure 6 pharmaceutics-13-01867-f006:**
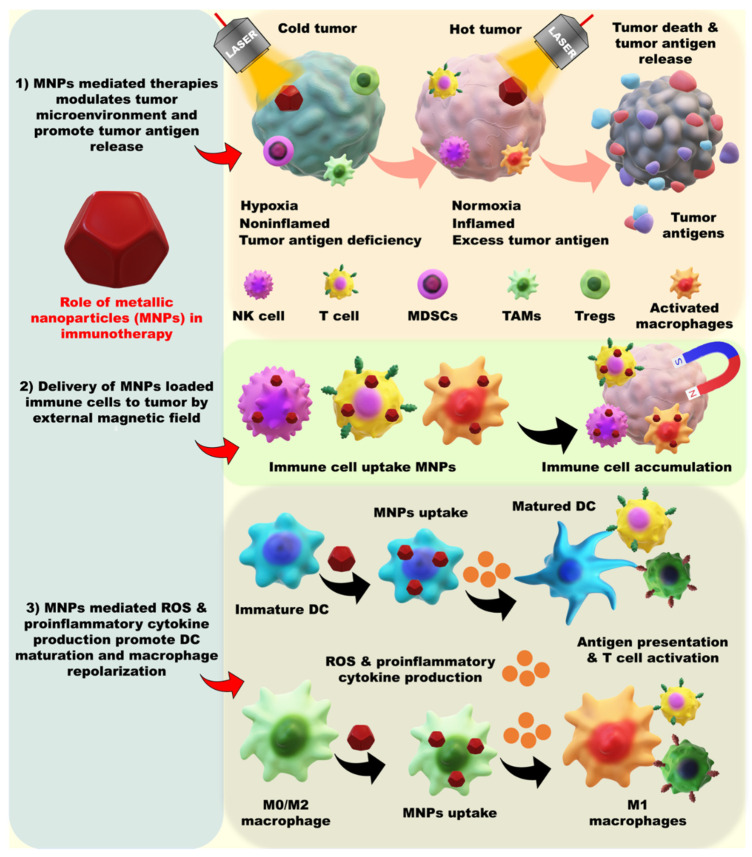
Schematic representation of the role of MNPs in immune cell modulation. (1) MNPs and MNP-mediated therapies modulate the TME conversion from cold to hot tumor type and represent accessible therapeutic strategies. MNP-mediated therapies, such as PTT, PDT, CDT, SDT, MHT, and IT, cause tumor cell death and promote tumor antigen production. (2) MNP-loaded immune cells, such as NK cells, T-cells and macrophages, are externally targeted to promote an immune response against tumors by external magnets. (3) MNPs promote intracellular ROS generation and pro-inflammatory cytokine production, which convert immature DCs to mature DCs and promote antigen presentation. Similarly, ROS and pro-inflammatory cytokine production promote pro-inflammatory macrophages conversion and elevate antigen presentation to cytotoxic T cells. MNPs stimulate intracellular pathways to activate antigen-presenting cells and induce a robust immune response against cancer.

**Figure 7 pharmaceutics-13-01867-f007:**
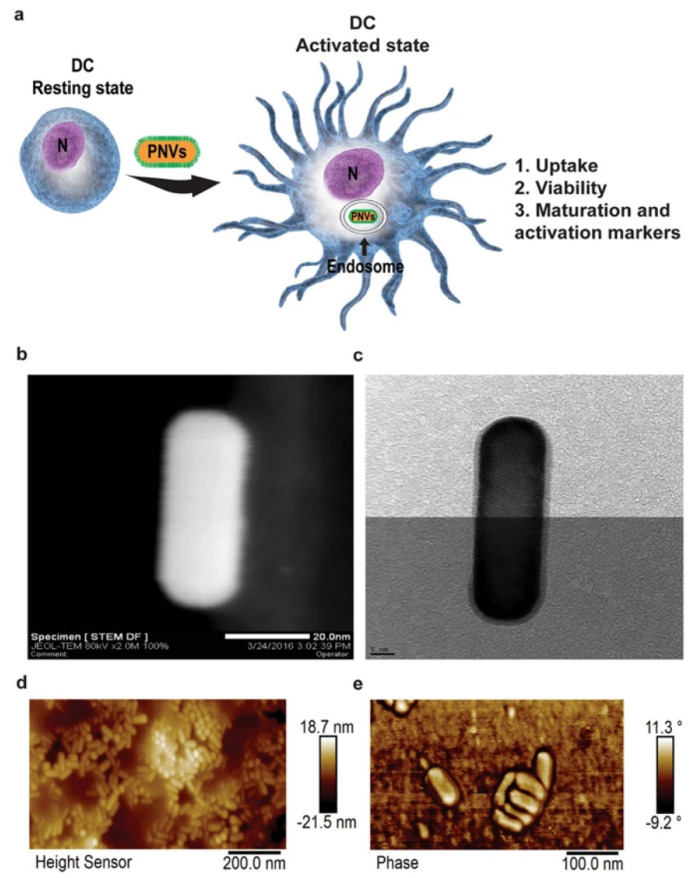
(**a**) Schematic diagram of a plasmonic nanovector (PNV) inducing DC maturation. Three factors, including PNV uptake, viability, and PNV-mediated DC activation marker upregulation, aid in the conversion of resting DCs into activated DCs. N indicates nucleus. (**b**) STEM and (**c**) TEM image of PNV. AFM image (**d**) left side and (**e**) right side of PNV. Reproduced from Vang et al. [94] which is licensed under a Creative Commons Attribution-(CC BY 4.0) International License (http://creativecommons.org/licenses/by/4.0/).

**Figure 8 pharmaceutics-13-01867-f008:**
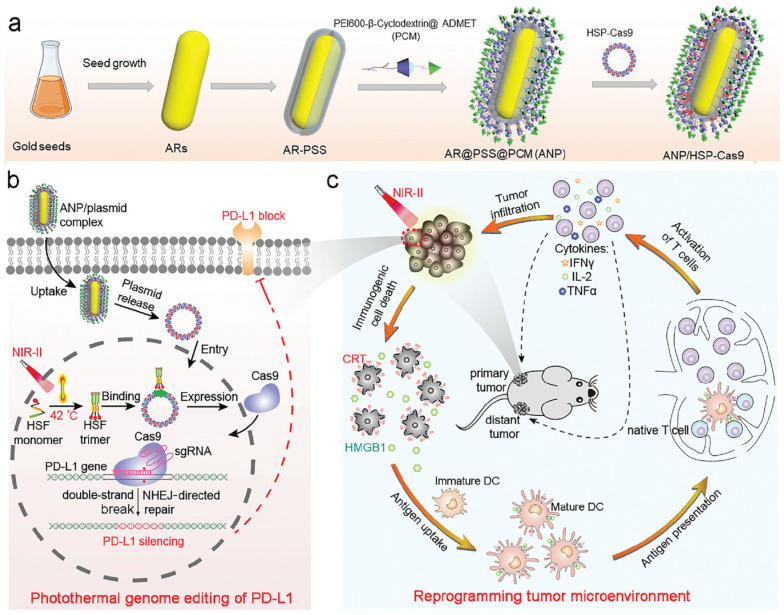
A schematic diagram of MNP-mediated ICD, tumor antigen release, and DC maturation. (**a**) Synthesis of the ANP/HSP-Cas9 complex. (**b**) Photothermal activation of PD-L1 genome editing. (**c**) NIR laser-induced photothermal therapy triggers ICD and antigen presentation, thereby activating the T cells. Reproduced with permission from [20] published by John Wiley and Sons, 2021.

**Figure 9 pharmaceutics-13-01867-f009:**
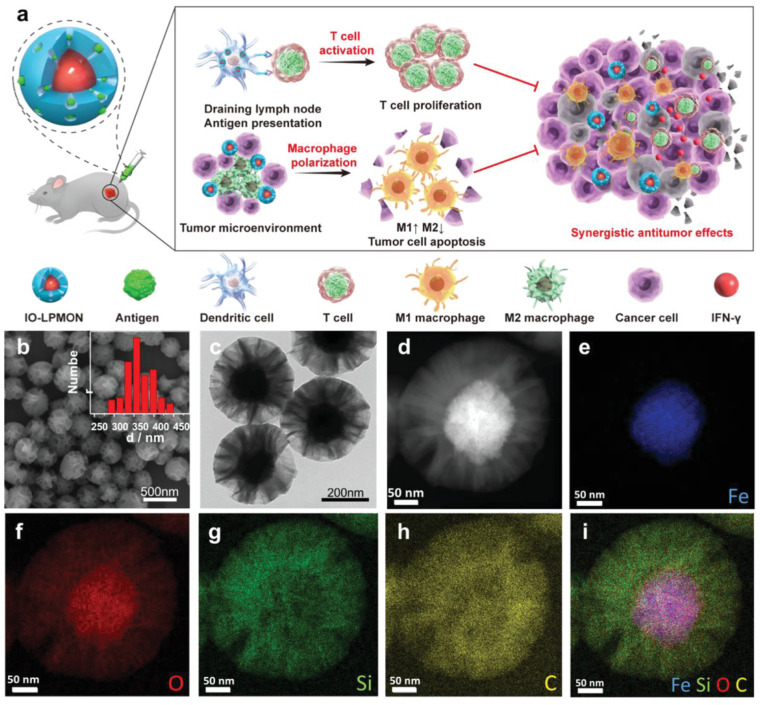
Schematic diagram representing MNP-mediated macrophage polarization for antitumor immunotherapy. (**a**) OVA antigen-loaded iron oxide nanoparticles modulate M2 macrophage polarization to M1 macrophages and stimulate T-cell proliferation. (**b**–**i**) Physiological characterization (TEM, SEM and elemental mapping of the IO-LPMON). Reproduced with permission from [106] published by John Wiley and Sons, 2021.

**Figure 10 pharmaceutics-13-01867-f010:**
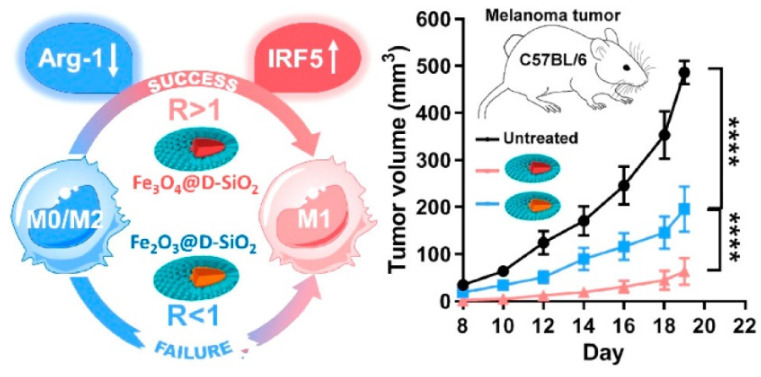
Schematic diagram of MNP-mediated macrophage polarization. The composition IONPs promote ratio (R) of intra cellular accumulation and elevateM0/M2 macrophage polarization to M1 and antitumor response. One-way anova was used for stastical analysis and results are displayed as mean ± SD, where **** *p* < 0.0001.

**Figure 11 pharmaceutics-13-01867-f011:**
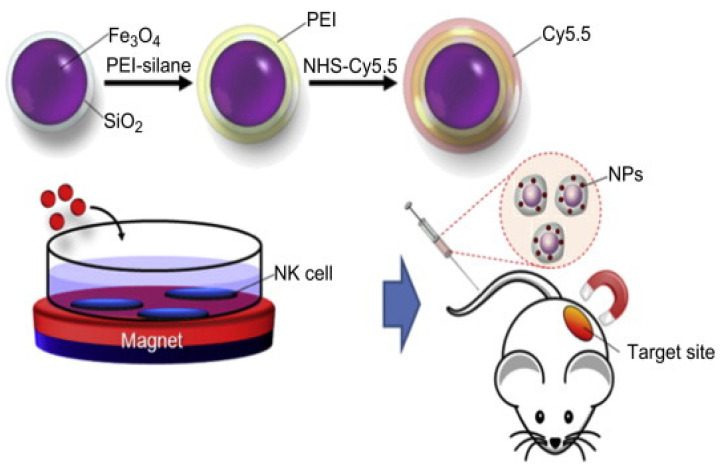
Schematic illustration of MNPs loaded in NK cells delivered to the tumor site and targeted with an external magnetic field. Reproduced with permission from [115] by Elsevier, 2012.

**Table 1 pharmaceutics-13-01867-t001:** Role of MNPs in TME modulation.

Nanoparticle Formulation	Therapeutic Action	Properties	References
MnO_2_	Hypoxia relief	Catalyzes intratumorally H_2_O_2_ and generates O_2_	[66]
MnO_2_ + GoX	Hypoxia relief	Reduces glucose content and improves oxygen availability by catalyzing H_2_O_2_	[67]
MnO_2_	Hypoxia relief	Improves the therapeutic efficacy by increasing oxygen content	[63,68]
MnO_2_ + Fe_3_O_4_/SiO_2_	Hypoxia relief	Oxygen boosters; release hypoxia by degrading H_2_O_2_	[69]
Pt nanoparticles + Zirconium shells	Hypoxia relief	Reduce tumor hypoxia and convert O_2_ into cytotoxic ROS	[70]
Pt-CuS Janus nanoconstruct	Hypoxia relief	Regulates the catalytic activity using Pt and improves the efficiency of sonodynamic therapy	[71]
Pt+ self-assembled micelle using Ce6 and PEG along with UCNPs	Hypoxia relief	Increases oxygen production and effectively generates ROS upon exposure to a 980 nm laser for tumor clearance Photo-chemotherapy of the tumor hypoxic environment	[72]
Fe_2_O_3_ + SiO_2_ and Au_2_O_3_	Hypoxia relief	Improves the anticancer effects of dox by modulating tumor hypoxia via light induced O_2_ production	[73]
CuO @ ZrO_2_coreshell	Hypoxia relief	CuO in the core shell ameliorates tumor hypoxia by improving oxygen level and boosting chemotherapy	[74]
Iron Oxide	Thermal ablation	Passive heat production for improved eradication of tumor microenvironment by inductively coupled plasma and AMF	[75,76]
Gold	Thermal ablation	Thermal ablation was achieved by delivering shortwave radiofrequency in order to destroy the tumor cells	[77,78]
Gold	Thermal ablation	The photothermal ability of internalized gold nanoparticles has been used to synergistically eradicate cancer cells	[79]
Gold nanostars	Thermal ablation	Exhibit improved photothermal ability upon internalization into endosomes both in vitro and in vivo	[80]
Silver Hybrid nanocomplex	Thermal ablation	Upon irradiation with an 840 nm laser, the hybrid nanocomplex was found to exhibit an increase in temperature levels, leading to cell death	[81]
Palladium	Thermal ablation	PDT/PTT combination therapy is effective in reducing tumor size compared with single therapy	[82]
Gold-silver nanocage	ROS generation	Owing to excessive production of ROS, the nanocomplex destroys the cell membrane, leading to apoptosis	[83]
MgO	ROS generation	Aids in lipid peroxidation and leads to apoptosis	[61]
CuO	ROS generation	Cell death occurs due to the increased production of ROS	[84]
ZnO	ROS generation	Varying concentrations of ZnO increases the levels of various ROS, leading to cell death	[85]
ZnO	ROS generation	Combined anticancer and antibacterial activity of ZnO nanoparticles via ROS generation	[86]
MgO	GSH depletion	Helps reduce GSH concentration in the tumor cells and aids in tumor destruction	[87]
MnO_2_	GSH depletion	Depletes the intracellular levels of GSH, thus improving the efficacy of chemodynamic therapy	[54]
Cu-TCPP MOF	GSH depletion	Efficiently degrades the intracellular GSH and converts it into oxidized glutathione	[59]

**Table 2 pharmaceutics-13-01867-t002:** Role of MNPs in immune cells.

Nanoparticles	Source Cells	Immune Responses	References
MnO_2_	Tumor-associated macrophages, CD4^+^ T helper cells, CD8^+^ cytotoxic T-cells	Release of tumor-associated pro-inflammatory macrophages and activation of T-helper cells and cytotoxic cells in order to initiate an immune response	[118]
MnO_2_	M1 macrophages	Significant reduction in M2 macrophage population and increase in M1 phenotype	[107]
Mn^2+^	DCs, T cell, NK cells, Macrophage	Stimulates STING activities with STING agonists and promote DC maturation, T cell activation, NK cell activation, and macrophage polarization	[95,96]
Ni^2+^	Human Toll-like receptor activation	Ni^2+^ selectively activates human Toll-like receptors to induce an immune cascade	[119]
TiO_2_	Toll-like receptors	TLRs activate macrophages and aid in immune cascade reactions	[120]
Fe_3_O_4_/SiO_2_	NK-92MI cells	The movement of NK-92MI cells can be controlled using a magnetic field, thus recruiting additional cells to the tumor site	[115]
Iron oxide	T-cell activation and cytokine secretion	Introducing iron oxide nanoparticles enhanced T cell activation and cytokine release	[121]
Silver	Cytokine inhibition	Introduction of silver nanoparticles into the tumor environment reduces the secretion of IL-1β (tumor-promoting cytokine)	[122]
Iron oxide	Pro-inflammatory macrophages	Iron oxide induces immune response by increasing the polarization of M1 pro-inflammatory macrophages	[123]
Iron oxide	CD8^+^ T cells	Mild hyperthermia in the tumor region improved the activation of dendritic cells and facilitated the entry of CD8^+^ T-cells in the draining lymph node	[124]

## Data Availability

Not applicable.

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
