# Peer review of "Metallic Nanoparticle-Mediated Immune Cell Regulation and Advanced Cancer Immunotherapy"

_pharmaceutics, 2021, doi:10.3390/pharmaceutics13111867_

Round 1

Reviewer 1 Report

This review paper regards an important area of cancer drug treatment. It is well documented. The overall content is well written. 

Author Response

Response to Reviewer-1

This review paper regards an important area of cancer drug treatment. It is well documented. The overall content is well written. 

Response-

We would like to thank the reviewer for the careful and thorough reading of this manuscript. We express our sincere gratitude to the reviewer for his approval of the manuscript. 

Reviewer 2 Report

Manuscript Reference No.: pharmaceutics-1435166

Title: Metallic nanoparticle-mediated immune cell regulation and advanced cancer immunotherapy

            In this manuscript authors have discussed in details about the different ways of how cancer bypass the immune system followed by application of metallic nanoparticles in the immune system activation and its mechanism. Also, various drug delivery strategies for modifying the cancer cell immunity with the help of metallic nanoparticles are also discussed. After reviewing this manuscript, I recommend the following changes.

  1. Need to discuss about the cited research articles in more detail for the concept clarification. For example, in line no. 340 authors have written the detail about one research articles; that should be in more details for concept clarification.
  2. Authors can write details about the interaction of metallic nanoparticles with biological system upon systemic delivery of nanoparticles.
  3. Various routes of delivery and their associated consideration is a missing topic which come under this review title.
  4. In line no. 65, authors have written as “nanoparticles lack” which need grammar correction as “nanoparticles have lack of”.
  5. In line no. 73, authors have mentioned the word “biological fluid pressure” which need to be changed either by biological fluid or interstitial fluid pressure.
  6. Title of this article include two main keyword “Metallic nanoparticles” and “Immunotherapy”, so some details regarding the nanomaterials is need to be include.

Author Response

Response to Reviewer-2

Comment-1

Need to discuss about the cited research articles in more detail for the concept clarification. For example, in line no. 340 authors have written the detail about one research articles; that should be in more details for concept clarification.

Response-

We would like to thank the reviewer for the careful and thorough reading of this manuscript and for the thoughtful comments and constructive suggestions, which help to improve the quality of this manuscript. According to the reviewer's suggestion, we have modified the highly positive and generalized statements.

In Section 4.1 and line number 361

MnO2 NPs were synthesized on the thiol functionalized mesoporous silica NPs (MS@MnO2 NPs) through thiol metal interaction. MnO2 depletes GSH to GSSG and produces Mn2+ ions through a redox reaction. These Mn2+ ions react with the intracellular H2O2 and bicarbonate to produce hydroxyl radicals, which enhance the intracellular production of toxic hydroxyl radicals (.OH). Mn2+ mediated GSH depletion prevents .OH radical scavenging and promoting enhanced CDT [54].  Similarly, Yang et al. demonstrated that iron oxide induces ROS via the Fenton reaction in the presence of intracellular H2O2 [17]. However, intracellular H2O2 level is not sufficient to process the Fenton reaction and hydroxyl radical generation. A concentration of 20µg/ml MNPs treatment in the HeLa cell line is insufficient to promote ROS-mediated cell death. Thus, exogenous H2O2 was required to promote Fenton reaction mediated CDT. The increased levels of ROS induce the secretion of pro-inflammatory cytokines at both the tumor site and in the systemic circulation. IL-6, IL-1β, and TNF-α are secreted at high levels in and around the tumor environment after the exposure of MNPs which alters the TME and improves immune response. Nanoparticles facilitate the electron transfer process and reduce the mitochondrial membrane potentials, thereby increasing ROS accumulation. Various other metal nanoparticles, such as Fe3O4, CuO, and MnO2, can produce a ROS burst inside the TME via Fenton and Fenton-like reactions [55]. MNPs mediated ROS production induce apoptosis and tumor-associated antigen (TAA) release. These TAA productions inside TME promote DC maturation and T cell activation against the tumor. A combination of CTLA-4 blockade therapy controlled primary and metastatic tumor growth [56].

In section 4.2 and line number 405-

Tumor cells carry high levels of GSH, as they need to scavenge the ROS that is repeatedly generated by the increased metabolic activity. Excess GSH level in cancer cells scavenges the singlet oxygen and limits the therapeutic efficacy of PDT. MNPs in the TME reduce the concentration of GSH via the formation of a metal-GSH thiolate, which in turn reduces GSH concentration inside the cell [58]. The Cu-TCPP complex enables the reduction of GSH to GSSG (a disulfide inactive form) via a single-step reduction of Cu2+ ions to Cu+ ions [59]. Cu-TCPP complex depleted the GSH level from 76% to 25% and increased the GSSG level from 24% to 75% in 5h. ESR spectrum intensity of 1O2 detection was reduced instantly after the addition of GSH but it was recovered after 6h which confirmed the GSH depletion mediated ROS generation. Intracellular 1O2 detection was performed through DCFH-DA assay in HeLa cells. The fluorescence intensity of DCFH-DA in HeLa cells was significantly reduced after the addition of GSH promoters. However, the fluorescence intensity of DCFH-DA was unchanged after Cu-TCPP treatment which was suggesting the role of Cu-TCPP mediated GSH scavenging and ROS production in cancer cells [59]. The decline in GSH concentration of tumor cells in response to the presence of metal nanoparticles induces the generation of excess ROS at the tumor site, thereby increasing inflammation and severe immune response [60,61].

In section 4.3 and line number 438-

MNPs are utilized in singlet and doublet oxygen generation and deliver the therapeutic loads into the site of action. Metals such as calcium, iron, copper, manganese, cerium, etc. can generate intracellular oxygen upon exposure to intracellular peroxides. Tingsheng Lin and his co-workers, in 2018, explained the mechanism of oxygen generation inside the TME (hypoxia) by MnO2. NPs. As shown in Fig.4, MnO2 reacts with the intracellular H2O2 to produce oxygen [63]. This catalytic reaction between intracellular H2O2 and MnO2. NPs increase the levels of intracellular oxygen, thus alleviating the hypoxic condition. Dissolved oxygen level was increased with the addition of H2O2, where a higher concentration of H2O2 and longer reaction time between MnO2 and H2O2 produced more oxygen. Similarly, the singlet oxygen production rate was higher in HSA-MnO2-Ce6 + H2O2 group than in the HSA-MnO2-Ce6 group. PDT effect of HSA-MnO2-Ce6 was significantly higher than HSA-Ce6. During in vitro cell viability assay, the HSA-MnO2-Ce6 group has shown 89.70 % cell death compared to 54.76% cell death in the HSA-Ce6 group. Moreover, in vivo tumor growth was suppressed significantly in the HSA-MnO2-Ce6 group due to the enhanced PDT effect [63]. Intracellular oxygen generation by MNPs is an emerging method to alleviate the hypoxic condition thereby, promoting the efficacy of other treatments.

Comment-2

Authors can write details about the interaction of metallic nanoparticles with biological system upon systemic delivery of nanoparticles.

Response-

We appreciate the reviewer for his valuable comment. As per the suggestion of the reviewer, we have included the summarized paragraph in the introduction.

Line number-52

The administration of nanoparticles into organisms can be carried out in many ways. Each route of administration has its advantages and disadvantages. Nanoparticles can be subjected into an organism by multiple routes such as oral, intramuscular, intravenous, intraperitoneal, and intranasal which eventually distributes the nanoparticles inside the body. Amongst these, the pulmonary drug delivery system shows a higher degree of drug delivery to the target organ but on the other hand, it also exhibits potential local toxicity leading to a critical and narrowed use of drug delivery [3,4]. Intravenous and oral drug delivery has better bioavailability and reduced systemic toxicity. Nanoparticles on systemic injection circulate in the blood and accumulate in various organs of the body. The major accumulation of the nanoparticles is traced in the liver. The liver acts as a detoxifying organ that helps in the removal of foreign and hazardous materials out of the system immediately. The nanoparticles are exerted out of the body in usually two ways such as renal excretion and hepatobiliary ways. Elimination of nanoparticles within a particular timeframe from the body stands out to be one of the crucial necessities for clinical approval. The major driving forces of nanoparticle clearance are the surface chemistry of the particle and the size. Smaller-sized nanoparticles provide better renal clearance. On the other hand, the surface coatings over the nanoparticle for example PEG coating helps in the extended circulation and always be inclined to have a hepatobiliary excretion [5].

Comment-3

Various routes of delivery and their associated consideration is a missing topic which come under this review title.

Response-

We appreciate the reviewer for detailed observation through our manuscript. We have included summarized a paragraph based on nanoparticle route of delivery, interaction with major organs, and their clearance from the body. The modified paragraph is included in the introduction part.

Line number-52

The administration of nanoparticles into organisms can be carried out in many ways. Each route of administration has its advantages and disadvantages. Nanoparticles can be subjected into an organism by multiple routes such as oral, intramuscular, intravenous, intraperitoneal, and intranasal which eventually distributes the nanoparticles inside the body. Amongst these, the pulmonary drug delivery system shows a higher degree of drug delivery to the target organ but on the other hand, it also exhibits potential local toxicity leading to a critical and narrowed use of drug delivery [3,4]. Intravenous and oral drug delivery has better bioavailability and reduced systemic toxicity. Nanoparticles on systemic injection circulate in the blood and accumulate in various organs of the body. The major accumulation of the nanoparticles is traced in the liver. The liver acts as a detoxifying organ that helps in the removal of foreign and hazardous materials out of the system immediately. The nanoparticles are exerted out of the body in usually two ways such as renal excretion and hepatobiliary ways. Elimination of nanoparticles within a particular timeframe from the body stands out to be one of the crucial necessities for clinical approval. The major driving forces of nanoparticle clearance are the surface chemistry of the particle and the size. Smaller-sized nanoparticles provide better renal clearance. On the other hand, the surface coatings over the nanoparticle for example PEG coating helps in the extended circulation and always be inclined to have a hepatobiliary excretion [5].

Comment- 4

In line no. 65, authors have written as “nanoparticles lack” which need grammar correction as “nanoparticles have lack of”.

Response-

We appreciate the reviewer for detailed observation through our manuscript. We have corrected the grammatical error.

In line 82-

However, lipid and polymeric nanoparticles have lack of long-term in vivo stability during systemic circulation and multifunctional applications.

Comment 5-

In line no. 73, authors have mentioned the word “biological fluid pressure” which need to be changed either by biological fluid or interstitial fluid pressure.

Response-

We appreciate the reviewer for detailed observation through our manuscript. We have corrected the grammatical error.

In line 91-

Unlike lipid or polymeric nanoparticles, MNPs tend to be highly stable under interstitial fluid pressure.

Comment-6

Title of this article include two main keyword “Metallic nanoparticles” and “Immunotherapy”, so some details regarding the nanomaterials needs to be include.

Response-

We appreciate the reviewer for detailed observation through our manuscript. We have discussed the various nanoparticles and their application in cancer therapy in the introduction part. We have summarized the route of nanoparticle administration and their circulation in the body. In this review manuscript, we have focused on metallic nanoparticle and their role in the immune response.

Reviewer 3 Report

The manuscript is well structured and presented. I have one minor suggestion that the authors need to address before considering for publication. 

1. In the whole manuscript the authors have just shown one figure i.e. figure 9 with one of the study data. If possible for the authors, they may insert some more data figures 2-3 representative figures regarding the studies done with MNPs.   

Author Response

Response to Reviewer-3

In the whole manuscript the authors have just shown one figure i.e. figure 9 with one of the study data. If possible for the authors, they may insert some more data figures 2-3 representative figures regarding the studies done with MNPs.   

Response -

 We appreciate the reviewer for the valuable comment and as per the suggestion of the reviewer, we have included figure 7 and 10 which includes the study data.